# Barriers and Enablers for Physical Activity in Culturally Deaf Adults: A Qualitative Thematic Analysis

**DOI:** 10.3390/ijerph22050777

**Published:** 2025-05-14

**Authors:** Alex B. Barker, J. Yoon Irons, Clare M. P. Roscoe, Andy Pringle

**Affiliations:** 1School of Psychology, College of Health, Psychology and Social Care, University of Derby, Kedleston Road, Derby DE22 1GB, UK; a.barker@derby.ac.uk; 2Clinical Exercise and Rehabilitation Research Centre, School of Sport and Exercise Sciences, University of Derby, Kedleston Road, Derby DE22 1GB, UK; c.roscoe@derby.ac.uk (C.M.P.R.); a.pringle@derby.ac.uk (A.P.)

**Keywords:** deaf, deaf-led activities, culturally deaf, physical activity, sign language, social identity, accessibility

## Abstract

**Purpose:** Physical activity (PA) is vital for everyone’s health and wellbeing; however, there is, a paucity of research amongst culturally deaf adults. Especially, to understand the needs of deaf adults and how to get them involved in shaping interventions that would help deaf people to be physically active. The current study aimed to explore barriers and facilitators for engaging in PA amongst deaf adults. **Method:** Focus groups involving nine culturally deaf adults communicating using British sign language were conducted and analysed using reflexive thematic analysis. **Findings:** Barriers including physical barriers, lack of deaf spaces and deaf awareness, and a lack of personal motivations were identified. Enablers included group/social support, deaf-led activities and health and wellbeing awareness. The findings highlighted a strong deaf identity. **Conclusions**: Deaf adults face barriers due to spaces being made for hearing people, leading to feelings of social exclusion and a lack of spaces to engage in activity and socialise, despite being personally and socially motivated to engage in PA. Deaf identity should be considered when promoting PA to deaf adults. The current paper highlights research and practice implications regarding how to engage and work with deaf people to develop appropriate interventions.

## 1. Introduction

Physical activity (PA) is well-known to improve both physical and mental health; for example, PA is reported to improve cardiovascular health as well as cognitive function [1]. In the UK, it is recommended that adults complete regular PA involving 150 min of moderate exercise or 75 min of vigorous exercise, or a combination of the two, alongside developing strength and balance activities at least twice a week [2]. Similar guidelines for older adults (65+ years) emphasise PA for promoting good physical and mental health [3] and reducing social isolation [4]. Yet in the UK, PA participation levels remain a cause for concern, and groups who encounter the greatest inequalities often face the greatest barriers to PA participation [5]. This includes adults with hearing loss. Recent estimates suggest 53.2% of people with deafness take part in less than 30 min a week of physical activity and are classed as inactive compared to 15% of people without a disability [6]. Given the positive effect that sport has been shown to have on issues such as depression and anxiety [7], these numbers are alarming and warrant the search for a better understanding of the experiences and barriers keeping deaf people from participating in regular PA.

There are two prevailing definitions of deafness. The medical model of deafness refers to deafness as a deficit of sound and refers to those who identify with hearing society. To date, the literature has focused on adults with acquired deafness, exploring PA in adults who were hearing and have lost their hearing or have a hearing loss, mounting epidemiological evidence has shown adults with acquired deafness are less physically active [8,9,10,11] and exhibit greater declines in physical functioning [12,13,14]. Moreover, of the scarce data available, individuals with deafness might be at a greater risk of health problems [15]. Evidence from prospective observational studies demonstrated that physical function (e.g., balance, sit-to-stand) declines at a faster rate in adults with deafness compared to those without [16,17]. Additionally, it was found that deaf men are more physically active than deaf women, age and body mass index are not associated with PA and a great proportion of deaf adults are overweight [18]. The current evidence suggests that there is a significant lack of PA and people with deafness are particularly at the greater risk of ill-health due to barriers to taking part [19,20].

However, an alternative definition of deafness exists. Deafness can also be defined as a state of being, defining a group of people who share a perception of the world through different sensory input [21]. Further, this definition is often applied to people who are deaf at birth or in early childhood and refers to a unique social and linguistic group, or culturally Deaf adults [21]. Whilst we recognise that this group is traditionally identified with a capital letter ‘D’ to represent Deaf Pride, we acknowledge on-going debate about the applicability of the term and for the remainder of the paper use the inclusive term deaf, but with the distinction between medical deafness and cultural deafness [22]. Whilst hearing-centric society views deafness in line with the medical model of deafness, as a medical condition or disability in need of a cure or adjustment [23], deaf culture rejects the disability construct, defining deafness as a difference, preserving a linguistic, cultural, minority [24]. There is, however, a lack of research evidence on the experiences of culturally deaf adults in relation to PA and differences may exist between those who define themselves using a medical model and those who align with deaf culture. In line with Glickman’s Deaf Identity Model [25], there may be differences between those who identify as culturally hearing and have deafness and those who are culturally deaf, with deaf and deaf-hearing bicultural people reporting higher life satisfaction [26]. The evidence base around adults with deafness and hearing loss may not apply to culturally deaf populations. Nevertheless, the social model of disability [27] posits that deaf adults are not disabled by their condition, but rather disabled by a world that does not meet their access needs [23], this can certainly be seen in the lack of accessible PA spaces [5]. Rather than assuming the same difficulties exist for people with acquired deafness and culturally deaf adults, the subjective experiences of this group should be explored. Therefore, the current study sets out to explore and understand the experiences of barriers and facilitators of culturally deaf adults engaging in PA, and to recommend how to involve deaf adults when developing PA interventions.

## 2. Materials and Methods

In meeting the aims of this study, focus groups were conducted with nine culturally deaf adults. Building on the existing relationship with a local school for deaf children, we recruited our participants from the teachers or teaching assistants at the school. Participants were between the ages of 35–65 years. Focus groups took place at the Royal School for the Deaf Derby; this was selected to allow members of the deaf community to be in a location where they would feel comfortable talking. Focus group discussions consisted of questions around their experiences of PA. The items in the interview schedule were informed by collaboration with the researchers and members of the deaf community and piloted prior to data collection. Interviewers (researchers) practiced their technique and working with a British Sign Language (BSL) interpreter. These preparations were necessary to ensure that the researchers acquired knowledge and learnt specific etiquettes to establish trust with the deaf community, which was in line with best practice of working with deaf sign language users [28]. Questions explored what activities participants currently do, any they used to do and have stopped (with reasons), what would help them to be physically active and any activities they would be willing to try. The researchers also learnt about working with BSL interpreters. A member of the research team was a deaf bilateral hearing aid user, who was present at the focus group discussions and able to advise the research team.

This study was approved by the University of Derby ethics committee (ETH2324-3898), and we obtained informed consent from the participants prior to the focus group discussions. Deaf adults who were over 18 years old were recruited. A certified and experienced BSL interpreter interpreted between the researchers and participants during the group discussion, which was video recorded. Following the discussions, the recordings were transcribed and checked by the interpreter for accuracy.

Data were analysed using reflexive thematic analysis [29], using a social constructionist epistemological stance. The six stages of analysis were followed, firstly, familiarisation of the data through engaging with the transcripts, codes were then developed to capture related meaning within the data. Themes were then developed and revised through continued interaction with the data and discussions amongst the research team. The consistency and trustworthiness of the analysis were ensured by having two authors working on the themes and sub-themes independently and having discussions with all authors to confirm them. Additionally, the authors recognise a constructivist epistemological position, where they have actively co-constructed meaning with the participants’ data and generated the themes and sub-themes [30,31]. AB is an experienced researcher on the topic of deafness, while CR and AP are experienced researchers in the field of PA. Additionally, JYI is an experienced qualitative researcher. Themes were then fed back to participants to ensure accuracy in our interpretation.

## 3. Results

A total of nine culturally deaf adults communicating using British Sign Language took part in two separate focus group discussions, lasting between 45 and 60 min. There were four females and five males, who were aged between 35 and 65 years old. From the data, two themes and six sub-themes were developed, presented in Table 1. An underlying theme of deaf identity was also identified across both barriers and facilitators. A selection of excerpts for each theme and sub-theme are included to highlight participants’ perceptions and needs.

### 3.1. Barriers

Key barriers to taking part in PA were expressed by the participants and the following three sub-themes were identified: (1) physical barriers, (2) the lack of deaf awareness, and a lack of dedicated space for culturally deaf people.

#### 3.1.1. Physical Barriers

Age and ability were seen as a barrier for engaging in PA and participants used this as a reason for not engaging in activity, presenting a view that PA and clubs are just not for older people.


*‘I was involved in a deaf football team but now I am too old for that.’*


Participants acknowledged that as we age, health problems can become more prominent, presenting a barrier for PA, showing a need for age-appropriate PA within the community.

For others, there was a lack of time due to their busy lives and the costs involved in engaging in PA.


*‘I work as a teacher, if I have time at home I just want to relax.’*



*‘[I] didn’t want to go to the gym, go swimming, but this [is] mainly costly and the time.’*


As well as the location of activities, which are often far away from the community. Many older people rely on others or public transport, and this can make travelling to take part in PA difficult.


*‘My parents, them, my mum can’t walk, she struggles, she can’t travel independently, she can’t get the bus, so she’s stuck, so she relies on my dad to drive.’*


#### 3.1.2. Lack of Deaf Awareness and Deaf Spaces

Participants spoke about how they are trying to take part in physical exercise in hearing spaces, which are not accessible or inclusive to the deaf community. Participants spoke about going to gyms or classes and experiencing spaces where the music is too loud and making communication difficult or being in spaces where everyone is talking and feeling excluded.


*‘I went to the gym. I went there and there’s a bit of a club, but I feel like I’m left out because they’re all talking and that’s a massive barrier.’*


To try to get around this, some participants had spoken about using a friend or interpreter to help communication, but this presented more barriers as deaf people still felt ‘othered’ in this situation and not a part of the group, and this still presented difficulties to taking part such as having to pay attention to the interpreter whilst trying to take part in a yoga exercise. Deaf people were excluded and disabled by trying to take part in a hearing group, showing the need for deaf aware and deaf spaces. Participants spoke about how there is a lack of space specifically for the community due to the closure of the local deaf club.


*‘But for deaf, there’s nothing. There’s hardly anything there. It’s really gone.’*


This lack of space may present a barrier to the community. Similarly to how deaf people felt ‘othered’ by trying to take part in a class which was not inclusive for them, a motivator for taking part in PA is the community and group-feel that people experience when the activity is inclusive or when they have a friend. Social Identity theory [32,33] states that we gain a sense of self-esteem and identity when part of a social group made up of similar others and engage in activities in line with the identity. Having a deaf space such as a deaf club can act as a motivator for PA, as can be seen in participants reminiscing about the use of the space in the past.


*‘Deaf sports you know, fantastic, really fantastic. I remember when I was small I lived in Nottingham, at Deaf Club and I’d come all the time and I’d see trophies lined up on the walls, challenges in people, deaf professionals, cricket, football, all sorts of sports and I would look and think that’s something I want to get involved in but now deaf clubs are reduced.’*


### 3.2. Enablers

Participants expressed what could help them engage with PA. Three sub-themes around enablers were identified: (1) group/social support, (2) personal motivations, and (3) Deaf-led activities.

#### 3.2.1. Group/Social Support

Having social support and being a part of a group was seen as an enabler to taking part in PA.


*‘And having a partner who’s going to go with me, you need a gym buddy. You need someone to like, bring you along and encourage you to go. That’s what you need.’*


Social support from a friend was seen as essential for taking part in activity, and participants often spoke about how their friend was often deaf and/or interpreting for them.

Spaces which were inclusive and accessible were seen as a positive light and removed the barriers for attending.


*[When discussing an inclusive group] ‘Some deaf and some hearing are there, but they both sign and it’s quite nice to have that group and it’s very relaxed to do that.’*


In addition, there was a strong desire that PA groups or activities should be deaf-led by the community, for the community.


*‘Obviously, they’d have to be deaf. You can’t have one deaf with an interpreter [it] has to be [a] group of deaf individuals.’*


#### 3.2.2. Personal Motivations

Other participants spoke about how sport and PA help with their mental health, providing stress relief, whilst also providing motivation and that connection to others.


*‘So, sport really helps, it’s really good for your mental health as well, connectivity [wise].’*


Other motivations included personal motivations for taking part in physical exercise, these involved wanting to stay fit and having goals, with one participant saying how they were aiming to fit into a dress they had.


*‘Before I went swimming in because that was my aim to, you know, wanted to wear that dress or I’ve got to be in that size 10 …You need something to aim for.’*


#### 3.2.3. Deaf-Led

Participants were aware of good practice in this area: for example, a local community group ‘Deafinitely Women’, which is a deaf-led group for women providing deaf-led PA programmes. This was praised by participants and perceived to be a group by them, for them.


*‘So, Derby Local Organization, Deafinitely women, have you heard of that before? Yes. They’ve got some funding to set up different activities, a small cohort of women that come together and get involved. They do lots of physical activities. They’ve started that this summer and they’ve got a programme, so walking, yoga, outdoor activities, lots of different things.’*


Similarly, other initiatives for the deaf community were seen positively for them, such as Derby Theatre, which was praised for using integrative theatre with a mix of hearing actors and signing actors. Participants noted how there have been increased attempts at inclusivity in different spaces, such as using subtitles, but felt that signing is best, and this related to the idea of the club being deaf-led, for deaf, by deaf.


*‘So definitely the sign language best. Yeah for me as well.’*


#### 3.2.4. Deaf Identity

Deaf identity was strongly presented in the data: particularly, belonging to the deaf social group was a significant condition for social, physical activities and an enabler for PA activities. The use of sign language was perceived as a significant marker of the deaf identity allowing participants to feel part of the group. Participants spoke about their deaf identity and how they identified as different from the hearing community and having different needs, and because of this, and the lack of deaf aware spaces, they required a deaf space for PA.


*‘It’s something they only provide for people who are hearing really they need to have something for the deaf community because we’re different…we have different needs…we’re culturally different.’*


## 4. Discussion

The current research explored the culturally deaf adults’ needs and experiences with PA, which has not been fully understood in the literature. The findings highlight barriers and enablers to PA for deaf adults. Through focus group discussions, key barriers to PA for deaf adults were identified; these barriers included physical difficulties regarding accessibility, and a lack of deaf-dedicated spaces where deaf people felt they belonged and able to join in PA. The study also found enablers which can help people engage in PA, such as dedicated group/social support for deaf adults, deaf-led activities and the aspiration to improve health and wellbeing status through PA. Further, this research found that a person’s sense of identity, specifically their deaf identity, was influential in a person’s willingness and ability to engage in PA.

In line with social identity theory, a strong sense of social identity can bring a sense of belonging, which is strongly associated with personal wellbeing. Behaving in congruence with and identifying with, a group can have a positive effect on self-esteem [31]. This study identified that spaces which are not deaf aware or deaf inclusive meant participants who felt ‘othered’ and not a part of the overall group due to being deaf. Without deaf-dedicated space, provision for deaf-aware PA provision, it is difficult for deaf adults to engage with PA because they do not feel they belong. Further, participants also spoke about how being in a group which is both deaf aware and inclusive, acts as a motivator for taking part in the group’s behaviour, in this case, PA. People can derive a sense of identity from the groups that they see themselves as part of, defined as social identity [32], with a person’s social identities contributing to a person’s overall sense of self [33,34], contributing to person’s mood and behaviours. Indeed, deaf identity has been defined as ‘Deafhood’, a process of deaf people navigating through their environments and experiences whilst constructing their own deaf identity [35].

Whilst deaf identity can be very personal and individual, the shared identity and experiences within the deaf community can allow a sense of belonging and allow the exchange of social support [26] in deaf friendly settings, such as deaf clubs [36]. This is in line with the deaf-led literature on deaf sports, participating in deaf sports can lead to gaining cultural bonds with a group and providing a space to socialise, enhancing self-esteem and community cohesion [37,38,39], whilst fostering and maintaining cultural identity among deaf people [40]. However, modern society is hearing-centric, defining deafness as defective hearing and in need of a cure [24,41,42], and as such, activities in hearing society are not designed with deaf people in mind. Instead, deaf people are disabled by a society that is not designed for them, where they need to adapt to fit in through accessibility measures [43], which discounts the deaf identity and the benefits of this. Furthermore, in line with previous research, the deaf community faces significant barriers to inclusion, such as the lack of sign language communication access and the lack of deaf coaches and deaf-led activities [37,38,44]. This has implications for improving PA in deaf adults; as such the involvement of deaf people in shaping public health solutions is key. Creating deaf-led group activities or groups which are deaf inclusive will enable a shared group identity, leading to engagement in PA. Indeed, according to the Inclusive Fitness Initiative, only 68 out of 7800 health and fitness clubs in the UK are accessible to disabled persons [5], and this does not mention those which are accessible to deaf people. Contrary to the view that deaf people are isolated and incapable of taking part in mainstream sports, when the provision is there, deaf people have been engaged in organised sport since the early 20th century [40]. Sporting clubs for deaf people have existed since 1888; however, despite the recent 100th year of the Deaflympics, there is a lack of visibility and differences in funding models for deaf sports [45], which may affect the perception of the availability of deaf-led PA. There’s also differences in whether sports and organisations are perceived as being for members of the deaf community, with specific examples including the rejection of the Paralympics by the deaf community due to a lack of alignment in self-determination, politicisation and activism [46], and the historical and social barriers leading to an underrepresentation of female deaf athletes [47]. Whilst we know that PA in older adults may have a positive effect on their health [48], there are potential barriers and facilitators to accessing PA [49]. Research facilitating dialogue with deaf people over their preferences and needs for PA participation and intervention, is very important in shaping effective solutions that promote PA [49].

Participants also spoke about a lack of deaf space, social spaces in which deaf people felt welcomed and valued. Participants lamented the loss of the local ‘deaf club’ where activities for the deaf community took place and where the Deaf community could socialise. Without this dedicated space/venue, participants felt there was nowhere for them to engage in activities, tied to the concept of hearing spaces not being for them due to a lack of accessibility, presenting a strong physical and social barrier to engaging in PA. Familiar places have been identified as being important for the promotion of PA in adults [50]. Deaf-led activities in a place which can be accessed by the community would allow the deaf community to feel part of the activities in a place where markers of their identity, such as the use of signed languages, are valued.

Another key finding of the current study is in relation to individual motivations to engage in PA involving setting goals for their physical fitness and maintaining their psychological wellbeing. This was in line with previous research, where participants who were aware of the positive effects of PA on their health were likely to engage in these activities [7]. Highlighting that raising awareness for a range of benefits from PA is essential for the deaf community. Again, the majority of health-related information is not inclusive, deaf adults may not be able to access vital information. Also, co-producing health-related information with the deaf community might be important, as such information will have greater meaning to them. Another motivator for engaging in PA was the opportunity for social connection and engagement with others, in line with developing a shared sense of identity and connection, this in itself acts as a powerful motivator for engaging in PA, in line with previous research [33,34]. Research has shown that social support is an important facilitator in PA participation with adults [49], including those adults with long term conditions and disability [51,52,53].

There were barriers to engaging in PA, participants spoke about physical barriers to engaging with activities, such as their self-perception of age and appropriate activities, as well as time and money to engage in behaviours. Another physical barrier was around the location of the activity and being able to access this, factors which need to be considered when we look to improve PA in the deaf community.

These findings and the interpretation of such need to be reflected on considering the study’s strengths and limitations. To ensure the quality of the research data, data were gathered using a registered British Sign Language interpreter, with data fed back to participants to check for meaning; however, transcripts were developed by the research team using the voice over of the interpreter from the recordings. In future, developing the transcript with the assistance of an interpreter could provide greater confidence in the accuracy. Data were then rigorously analysed among the research team. Once themes were drawn out, with the support of quotations, these were discussed and developed amongst the research team, which included a deaf academic. The study provides candid insightful accounts from the perspective of deaf adults using an inclusive facilitatory approach. These provide valuable insights that can help inform future PA provisions that meet the needs of this group. Further, this study also provides valuable information on how to conduct research in this context in partnership with Deaf people that can be valuable for other researchers engaging in this space. This is especially important given the significant investment that is being made in community PA for local populations whose PA needs are not being met, including those using bottom-up, appreciative and iterative approaches to better understand the multi-layered inequalities these groups face when becoming active [54]. Further, our findings will be shared through our regional PA networks [55,56], several of which are centred on addressing inequalities to PA participation. The current study involved a small number of unique participants at a certain time, in a single geographical location, speaking in a certain context, and this should be considered in the transferability of results and conclusions, specifically, whilst there is a lack of provision and accessibility for the subjects of the current study, we do know that deaf-led sporting groups do exist around the country. Future studies should consider participants who are accessing sporting groups.

## 5. Conclusions

This study has provided in-depth insights into culturally Deaf adults’ experiences of and needs for engaging with PA. Deaf adults face significant barriers due to inaccessible spaces, leading to feelings of social exclusion. For Deaf people to engage in PA, deaf identity is the key factor, that could personally and socially motivate deaf people.

This paper identifies a number of practical implications for working with Deaf adults to improve their PA: (1) Conduct and engage in regular dialogue with Deaf adults; (2) use accessible, adaptable, and inclusive approaches that help participants to share their account of PA characteristics that impact their needs; (3) deploy their perspectives to inform the discussions that shape interventions with stakeholders about meeting their needs; (4) involve Deaf adults in discussions with provider when designing PA interventions that best meet their needs; and (5) the dialogue should be ongoing throughout.

Future studies should also explore how best to raise health and wellbeing awareness for the Deaf community by engaging with the Deaf community and how Deaf-led approaches could enhance PA using robust methodologies, such as drawing on examples where this is currently working in established sport groups.

## Figures and Tables

**Table 1 ijerph-22-00777-t001:** Themes and Sub-themes.

Themes	Sub-Themes
Barriers	1.1 Physical Barriers 1.2 Lack of deaf awareness and space
2. Enablers	2.1 Group/social support 2.2 Personal Motivations 2.3 Deaf-led activities 2.4 Deaf Identity

## Data Availability

The data presented in this study are available on request from the corresponding author due to ethical considerations.

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
