# Peer review of "Barriers and Enablers for Physical Activity in Culturally Deaf Adults: A Qualitative Thematic Analysis"

_ijerph, 2025, doi:10.3390/ijerph22050777_

Round 1
Reviewer 1 Report
Comments and Suggestions for Authors
Manuscript ID: ijerph-3562651
Title: Barriers and Facilitators for Physical Activity in Culturally Deaf Adults
Journal: Int. J. Environ. Res. Public Health
Overall Assessment
The article is an important study that aims to examine the barriers and facilitators to physical activity (PA) participation of culturally Deaf adults. The topic is under-researched in the literature and provides valuable data in terms of public health. The study addresses an under-researched topic in the literature. The methodology was designed to accommodate the needs of Deaf participants. The use of a BSL (British Sign Language) interpreter is positive. The role that Deaf identity plays in participation in physical activity is effectively emphasized. However, certain corrections and additions would improve the scientific quality of the article and its contribution to the literature. In particular, eliminating the deficiencies in the methodology section and explaining the analysis process more transparently would increase the credibility of the research.
Title and Abstract
The title may specifically state that the focus is on "culturally Deaf" adults.
Abstract Structure: The abstract could be developed to be more structured and have clear purpose, method, findings, and conclusion sections.
Keywords: "Cultural Deaf", "Sign Language", "Social Identity" could be added to the keywords.
Introduction
Lines 42-54; The difference between acquired deafness and cultural deafness could be explained more clearly
Lines 70-74; The purpose statement could be more specific and measurable.
Lines 55-64; The theoretical framework regarding the Definitions of Deafness could be presented earlier in this section.
Method Section
In terms of verifying the participants, it would be appropriate to explain the process of verifying the findings by the participants more clearly.
How were the focus group interviews structured? What were the questions? More details should be provided.
Lines 76-90; It would be appropriate to present the characteristics and demographic information of the participants in more detail. More detailed information such as age distribution, education status, profession etc. can be added
Lines 76-79; For Sample Selection, the sampling method can be explained more clearly.
Lines 83-85; Spelling Error; "These" should be written instead of "Thess"
Lines 90-92 and Line 329; The ethical approval number is stated in two different sections as “ETH2425-0166” and “ETH2324-3898”.
Lines 97-102; Within the framework of Data Analysis, more details on the Reflexive thematic analysis process should be provided. The analysis process should be explained in more detail. The coding scheme, theme development process and how reliability checks were performed should be explained.
Results Section
Lines 107-110; Themes: The theme of identity is not included in the table but is mentioned in the results.
Line 111; There is an inconsistency in Table 1: “Personal motivations” does not appear in the table as a sub-theme, but it is mentioned in the text.
Lines 209-214; In terms of Theme Consistency, the sub-theme “Deaf Identity” is not mentioned in the table but is presented as a separate sub-theme in the findings section. This inconsistency should be corrected.
In the findings section, participant quotes for some themes are limited; some sub-themes have only 1-2 quotes. Some quotes could be presented with more explanatory context.
The themes should be analyzed in more depth and the relationships between sub-themes can be shown more clearly.
Discussion Section
The link to the literature could be more extensive. For example, more literature reviews on Deaf culture and identity and a discussion of how these findings overlap with or differ from previous studies.
The theoretical framework (social identity theory) is well used in the discussion section, but other relevant theories could also be mentioned.
There may be a typo in line 232, revise again.
Lines 225-236; Within the Conceptual Framework, social identity theory can be presented with earlier references to the literature
Lines 250-253; Structural/systematic barriers of physical activity models can be discussed more comprehensively in terms of Linkage with Related Research
Lines 301-304; Limitations of the study regarding the generalizability of the findings can be discussed more extensively
Lines 310-314; Regarding Implementation Suggestions, practical suggestions can be presented in a more specific and applicable way.
Lines 315-318; This section on Future Research should be expanded and more specific research questions should be proposed.
Conclusion and Implementation Recommendations
Implementation recommendations could be more specific. Suggestions such as “Conduct and engage in dialogue” and “using accessible adaptable, and inclusive approaches” are good, but too general; more concrete suggestions could be developed.
Recommendations for future research are too brief and could be more detailed.
Bibliography and References
Web sources could be supported with more academic sources.
Lines 207-397; Some citations seem to be missing or incomplete, citations should be checked in this respect.
Spelling and Language Usage
Spelling Consistency: The use of "deaf" and "Deaf" should be consistent throughout the text.
Format: The spaces between subheadings are not consistent, they should be checked.
Line 52; "the" started with a lower case letter.
General Recommendations
Limitations of sample size should be discussed more clearly.
More theoretical explanations should be provided for why and how physical activity is important for deaf adults.
A clearer definition of a research gap should be provided in the introduction.
A general language check should be conducted on the article, and spelling and grammatical errors should be corrected.
More concrete suggestions should be developed for how accessible physical activity can be designed for the deaf community.
These corrections and suggestions will improve the scientific quality of the article and enable its findings to be used more effectively to support deaf adults’ participation in physical activity.
Conclusion
The article is an important study that sheds light on the barriers faced by the Deaf community in accessing physical activity. By making the suggested corrections and additions, the scientific quality of the article and its contribution to the literature will increase. In particular, eliminating the deficiencies in the methodology section and explaining the analysis process more transparently will increase the credibility of the research.

Author Response
Dear Sir/Madam,
We thank you for taking the time to review our manuscript and feel this has been strengthened by your contributions. In the attached document, we reply to your comments and detail the changes we have made in response to this. Comments are in black ink with our reply below in blue ink.
We genuinely thank the reviewers for their comments and appreciate the time taken to improve this work.
Kind regards
Dr Barker and team

Reviewer 2 Report
Comments and Suggestions for Authors
Nice paper with just a few comments about the writing in the beginning and my belief that you discussion needs to be rewritten to pull together an interesting history of deaf sports written MOSTLY by deaf people.
Please add member checking and validity to the procedure section of the paper--you can see I asked myself a question then found it much later.
Your discussion MOSTLY repeats your findings --what I really like about this paper is that you separate out the Deaf people from those who are deafened. This is important. Given the research you have this list of articles I found on Google Scholar in the attached file includes some from the UK that I did not see if you references.
I would like to see you weave these articles together and discuss how hearing people (an etic perspective) has assumed that Deaf are not involved in sports. These show the Deaf Olympics and other spaces.
You have captured the issue with Deaf Spaces but it would be refreshing to see a discussion that highlighted the emic (insider) perspective of those Deaf individuals you had in the focus groups!!
Reviewer's further comments:
The procedure section of the method needs to be expanded just a bit
more with more discussion of The Braun and Clarke method and
validity—-ie., member checking should go there as well as how they
checked each other and had a deaf author or researcher
What I really want to see if a rewrite of the discussion that
integrates their findings with the research that they did not
include—-they find the need for Deaf spaces. Deaf Olympics makes
that space—-people who are truly integrated into Deaf Culture know of
these events and some of the references I found even occurred in the
UK

Author Response

(The authors gave the same response as above.)

Round 2
Reviewer 1 Report
Comments and Suggestions for Authors
The authors have addressed the reviewer comments and made the necessary revisions in accordance with the review report.
Author Response
We thank the reviewers for their comments and are pleased to see they agree that the manuscript has been improved.
Reviewer 2 Report
Comments and Suggestions for Authors
Thanks for the revision
Author Response

(The authors gave the same response as above.)
